# Adverse Events of Vaccination against Hepatitis B Virus in Post-Marketing Surveillance from 2005 to 2017 in Guangdong Province, China

**DOI:** 10.3390/vaccines10071087

**Published:** 2022-07-06

**Authors:** Yu Liu, Minyi Zhang, Meiling Yang, Qing Chen

**Affiliations:** 1Department of Epidemiology, School of Public Health, Southern Medical University, Guangzhou 510515, China; wsjkw_liuy@gd.gov.cn (Y.L.); myalison@smu.edu.cn (M.Z.); 2Guangdong Provincial Center for Disease Control and Prevention, Guangzhou 511430, China; sjkzx_yangmeiling@gd.gov.cn

**Keywords:** hepatitis B virus, vaccine, adverse events, surveillance

## Abstract

The present study focused on the adverse events following the vaccination against hepatitis B virus (HBV) in the Guangdong Province of China between 2005 and 2017. In total, more than 88 million doses of HBV vaccine were administered in the Guangdong Province during the study period. A total of 3115 adverse events following immunization (AEFI) related to HBV vaccination occurred, with an overall incidence of 35.39 per million doses. Of these, 1801 cases were male, and 1314 were female; 74.01% (2376/3115) of the cases occurred in children aged less than 2 years; 56.05% (1746/3115) of the cases were classified as common vaccine reactions; and 30.37% (946/3115) of the cases were grouped into rare vaccine reactions. Additionally, 27.74% (864/3115) of the cases were classified as allergic reactions, 0.10% (3/3115) were temporary neurological events, and 1.28% (36/3115) were diagnosed as severe adverse events. This study suggested that the HBV vaccine posed a reasonable profile because most adverse events remained relatively mild, and the neurological events were relatively rare. This study concluded that the incidence of severe vaccine reactions related to HBV vaccination are extremely low.

## 1. Introduction

Hepatitis B virus (HBV) infection remains a critical issue of public health across the globe [1], which can lead to a lifelong chronic infection associated with the development of cirrhosis and hepatocellular carcinoma [2,3]. Hepatitis B virus is a partly double-stranded DNA virus with several serological markers, including HBsAg and anti-HBs, HBeAg and anti-HBe, and anti-HBc IgM and IgG [4]. It has been reported that nearly 30% of the population around the world indicates serological evidence of current or past HBV infection [5,6]. HBV infection frequently occurs through various transmission routes, including perinatal transmission, horizontal transmission (exposure to infected blood), percutaneous or mucosal exposure to infected blood, saliva, menstrual, vaginal, or seminal fluids, and sexual transmission [5,7,8]. Currently, roughly two billion people worldwide have been infected with HBV [9]. The most severe burden of HBV infection is found in China [10]. The National HBV Serological Epidemiology Survey in 1992 discovered that the prevalence of HBV infection was 9.75% among people aged less than 59 years, and it was up to 9.67% in children between 1 and 4 years of age. Subsequently, the National HBV Serological Epidemiology Survey in 2006 demonstrated that the prevalence of HBV infection had decreased to 7.18% among people aged less than 59 years, while it showed a sharp decline to 0.96% among children aged from 1 to 4 years [11]. Since then, it has been a medium-endemic area of HBV infection in China [12,13]. Hepatitis B passive and active immunoprophylaxis at birth, together with antiviral treatment of highly viremic mothers, are the key strategies for the global treatment of HBV infection [14,15]. The HBV vaccine for newborns is an essential measure to stop HBV infection and transmission from mother to their children. In 1992, the HBV vaccine was launched into the immunization program in China, and the free HBV vaccination had not been introduced until 2002 [16]. In light of the National Immunization Program Survey, the coverage rate of HBV vaccination was reported at 22% in 1992, and it had increased to 66.8% in 2002. However, it was slightly declined to 65% in 2014 [11]. The extensive promotion of the HBV vaccine plays an essential role in reducing HBV infection in China.

The adverse events of HBV vaccination have reached more and more attention from the general population and social media following its widespread application. In accordance with the HBV vaccination procedure for children, the first dose of HBV vaccine should be given immediately on birth, and the second and third doses of HBV vaccine ought to be given at the age of 1 and 6 months, respectively. Previously, a medical center in the Sichuan Province of China had reported 13 vaccine-related death cases among newborns from 2009 to 2014. Of these, 12 death cases were in relation to cardiovascular and respiratory diseases that were considered coincidental diseases following immunization [17]. Thus, there is a high likelihood of death of coincidental diseases following immunization of HBV vaccine among children aged less than 1 month [18].

Eighteen severe adverse events following immunization (AEFI) occurred at the end of 2013 in China after HBV vaccination produced by Shenzhen Kangtai Biological Products Co., Ltd. These included 17 deaths, and 6 of the 17 death cases were reported in the Guangdong Province [19]. Indeed, this event dramatically impacts the parental willingness toward HBV vaccination for their children. As a result of the investigation by the Ministry of Health (MOH) and the State Food and Drug Administration (SFDA), no causal association existed between the death events and HBV vaccination. The main causes of the 18 cases included severe pneumonia, asphyxia, renal failure, severe pediatric diarrhea, and congenital heart disease, etc. However, the public has questioned the quality and safety of the HBV vaccine despite the lack of association of deaths with HBV vaccination. Poor understanding of healthcare workers on adverse events related to vaccinations would reduce the recommendation of vaccination to the public [12]. To address this issue, we conducted the surveillance of adverse events in relation to the HBV vaccination between 2005 and 2017 in the Guangdong Province, China.

## 2. Methods

### 2.1. HBV Vaccine

In China, recombinant HBV vaccines have been widely applied since 2000. Currently, the recombinant HBV vaccine (Saccharomyces cerevisiae), recombinant HBV vaccine (Chinese hamster ovary cell), and recombinant HBV vaccine (Hansanula polymoupha) are used to prevent the population from HBV infection.

In the Guangdong Province of China, five recombinant HBV surface antigen vaccines were available to the public from 2005 to 2017. These included three recombinant HBV vaccines (Saccharomyces cerevisiae) manufactured by Shenzhen Kangtai Biological Products Co., Ltd., China-licensed (Shenzhen), Beijing Tiantan Biological Products Co., Ltd., China-licensed (Beijing), and GlaxoSmithKline Biologicals, U.K.-licensed, one recombinant HBV vaccine (Hansanula polymoupha) manufactured by Dalian Hissen Bio-pharm Lnc, China-licensed (Dalian), and one recombinant HBV vaccine (Chinese hamster ovary cell) manufactured by NCPC Genetic biaotecnology Co., Ltd., China-licensed (Shijiazhuang). The production standard of HBV vaccines made in China has been approved by the World Health Organization (WHO).

In regard to the vaccination schedule, newborns are given their first dose of HBV vaccine at birth, usually along with the BCG vaccine. Subsequently, the second and the third doses of the HBV vaccine would be applied at the age of 1 and 6 months, respectively. Children and adults that have never been vaccinated against HBV infection are recommended to receive the HBV vaccination. The vaccination schedule for these people is the 0th, 1st, and 6th month mark, and the HBV vaccine is administered alone.

### 2.2. Report System

With the large number of administrations every year and increasing attention on vaccine safety from the public and the media, even health workers in China are required to establish the monitoring of adverse events and to possess the capacity to rapidly respond to any novel emerging vaccine safety concerns. Monitoring guidelines for the surveillance of AEFI in accordance with the WHO have declared that AEFI refers to a reaction or an event after the immunization that is suspected to be concerned with the vaccination [20]. This is a passive pharmacovigilance system. Serious adverse events comprised hospitalization, death, life-threatening illness, and permanent disability [21]. Between 2005 and 2008, the Guangdong Center for Disease Control and Prevention (GDCDC) was responsible for implementing and reporting AEFI monitoring data through a self-developed report system by GDCDC. Subsequently, the National Online AEFI Report System was established in 2008 and was further applied for the surveillance of AEFI. An AEFI is any untoward medical occurrence that follows immunization, and that does not necessarily have a causal relationship with the usage of the vaccine. Healthcare facilities, vaccination units, and vaccine manufacturers are responsible for reporting an adverse event without a financial reward.

### 2.3. Report Scope

In light of the guidelines of the Chinese MOH and SFDA, the report scope of AEFI consists of the following categories based on the occurrence time [21]:Within 24 h: anaphylactic shock, allergic reactions without shock (urticarial, generalized rash, laryngeal edema, etc.), toxic shock syndrome, syncope, and hysteria psychosis.Within 5 days: fever (axillary temperature ≥ 38.6 °C), angioedema, systemic purulent infection (toxemia, septicemia, sepsis), redness and swelling in the injection site (diameter > 2.5 cm), scleroma (diameter > 2.5 cm), and localized purulent infection (localized abscess, lymphangitis, and lymphadenitis, cellulitis), etc.Within 15 days: measles-like or scarlet fever rash, Henoch–Scholein purpura (HSP), localized allergic necrosis reaction (Arthus’s reaction), febrile convulsion, epilepsy, polyneuritis, encephalopathy, encephalitis, and meningitis.Within 6 weeks: thrombocytopenic purpura, Guillain–Barré syndrome, and vaccine-related paralytic polio.Within 3 months: brachial plexus neuritis and sterile abscess in the injection site.Others: serious AEFI and any medical events in relation to the vaccination.

Some manifestations were classified as allergic reactions, including anaphylactic shock, allergic reactions without shock (urticarial, generalized rash, and laryngeal edema, etc.), angioedema, HSP, localized allergic necrosis reaction (Arthus’s reaction), etc. The neurologic reaction includes syncope, hysteria psychosis, febrile convulsion, epilepsy, polyneuritis, encephalopathy, encephalitis and meningitis, and brachial plexus neuritis, etc. Regarding AEFI, it can be categorized into common vaccine reaction, rare vaccine reaction, coincident reaction, psychogenic reaction, program error event, and unknown [22].

The above report scope of AEFI applies to the surveillance system and considers adverse events related to the vaccination but may not necessarily represent a causal relationship between the adverse events and the vaccinations.

### 2.4. Reporting and Investigation

As mentioned before, healthcare facilities, vaccination units, and vaccine manufacturers are responsible for reporting an adverse event. All these departments are obliged to report AEFI cases by using the same AEFI monitoring and reporting manual. When receiving a report of an adverse event, the reporting unit is required to fill out a case-report form and submits it to the county CDC. After verification, the county CDC should report it through the online National AEFI Surveillance System. Notably, the county CDC should investigate the adverse events and complete the detailed case-investigation forms for all adverse events. However, a detailed case-investigation form was not required for a mild adverse event with a clear diagnosis (e.g., fever, redness, and swelling on the injection site).

According to the guideline for the Identification of Adverse Reaction after Immunization issued by the Chinese MOH [23], each county, prefectural, and provincial CDC must set up an expert panel to investigate the adverse events using the criteria given by the Chinese Standard Procedures for Vaccination. Each panel consisted of physicians, epidemiologists, pharmacists, and other relevant experts. In general, prefectural or provincial expert panels investigate deaths, life-threatening illnesses, and permanent disabilities; county-level expert panels investigate other adverse events, and immunization program managers or vaccination providers investigate common, minor adverse events.

Information from the case-report form and detailed case-investigation form is reported to the GDCDC through the online National AEFI Surveillance System or Guangdong Province AEFI Surveillance System. All serious adverse events are investigated by expert panels of the county, prefectural, or GDCDC immediately after receipt of the reports. Prefectural and GDCDC check the accuracy and completeness of the data within 1 week after the adverse events are reported.

### 2.5. Data Analysis

We collected data on age, gender, the time interval from vaccination to the onset of adverse events, symptoms, and case diagnosis. Only the main symptom or the most severe diagnosis was recorded in the AEFI reporting system when one person presented more than one symptom. The outcome of this study included the rate of adverse events and the proportion of each event category. Crude rates of AEFI, serious AFEI, allergic reaction, and neurologic events following HBV vaccination were calculated per million doses. The information in this study is routine surveillance from GDCDC. The research protocol was approved by the ethics committee of Southern Medical University.

## 3. Results

A total of 88.02 million doses of HBV vaccine were administered in the Guangdong Province from 2005 to 2017. Of these, 3115 AEFI cases occurred, with an overall incidence of 35.39 per million doses. Among the AEFI cases, 1746 patients presented common vaccine reactions, 946 presented rare vaccine reactions, 342 coincidence reactions, 60 psychogenic reactions, 3 vaccination accidents, and 11 patients remain unclear (Table 1). The incidence of a common vaccine and rare vaccine reactions was 19.84 and 10.75 per million doses, respectively (Figure 1). Overall, the number of AEFI cases indicated an overall upward trend. The highest number of AEFI cases were reported in 2014. A significant increase in the incidence of AEFI in 2014 was noted. Among the AEFI cases, 1801 were male, and 1314 were female. The majority of cases occurred in infants aged less than one year (Table 2), while the lowest number of AEFI cases were reported in people aged greater than seven years.

Regarding the classification of adverse reactions, 1746 cases were grouped into common vaccine reactions, and the remaining cases were rare vaccine reactions (Table 3). The most frequent cases were diagnosed with fever among the common vaccine reactions (n = 1046), with an incidence of 11.88 per million doses, followed by redness and swelling (n = 545), and the minor clinical manifestation was nausea (n = 1) and headache (n = 1). Additionally, 946 cases were classified as rare vaccine reactions, with an incidence of 10.75 per million doses. More specifically, the primary rare vaccine reactions included anaphylactic rash (n = 716), with an incidence of 8.13 per million doses, followed by urticaria (n = 69), sterile abscess (n = 31), angioedema (n = 24), thrombocytopenic purpura (n = 21), and maculopapular rash (n = 20). The lowest incidence of rare vaccine reactions was reported at 0.01 per million doses in laryngeal edema, brachial neuritis, encephalopathy, and encephalitis and meningitis (all n = 1).

Psychogenic reaction is defined as an AEFI arising from anxiety following the immunization. In 2010, a total of 47 students aged between 9 and 15 years self-reported symptoms of dizzied and diarrhea when being vaccinated in a school of Guangdong Province. Following the investigation, they were diagnosed with psychogenic reactions without any clinical manifestation [24]. Only three program error events were reported; two error cases of purulent infection at the inoculation site were reported in 2009 and 2010, respectively. In 2006, an error case of local pain occurred due to repeated vaccination.

People aged over 7 years or those never vaccinated against HBV infection are recommended to receive HBV vaccination. Thus, it is possible that children aged over 7 years may be one of the reported AEFI cases. In the present study, a total of 333 adverse events occurred in children aged more than 7 years, and the most frequent types of reactions reported in this age group was redness (n = 124).

## 4. Discussion

This study analyzed the adverse events following HBV vaccination in the Guangdong Province from establishing the vaccine-related adverse events monitoring system from 2005 to 2017. Our findings indicated that the lowest incidence of AEFI related to HBV vaccination was reported at 4.52 per million doses in 2005, while the highest one was reported at 79.62 per million doses in 2014. In regard to the rare vaccine reactions, the lowest incidence was observed in 2005 at 2.35 per million doses, while the highest one was reported at 26.13 per million doses in 2014. Overall, the incidence of AEFI and rare vaccine reactions of the HBV vaccination showed an upward trend [25]. These were in relation to the continuous improvement of the AEFI monitoring system and the gradual improvement of monitoring sensitivity.

The quality of the Chinese national AEFI monitoring system has been continuously improving since 2005. The number of provinces implementing AEFI case reports has increased from ten pilot provinces in 2005 to all provinces in 2010, except the Tibet Autonomous Region, while the county-level coverage rate has increased from 12.74% in 2005 to 81.84% in 2010. Regarding the AEFI events, the number of cases has also increased from 1932 in 2005 to 54,663 in 2010 [26]. Likewise, 36,776 AEFI events due to vaccination were reported in Guangdong Province between 2005 and 2014, which also showed an upward trend year by year [25].

In the present study, a large number of AEFI cases by HBV vaccination were reported in children younger than one year (n = 2202, 70.69%). This might be in relation to the vaccination procedure for the HBV vaccine that is primarily applied to infants aged less than one year. Moreover, 1801 and 1314 (1.37:1) AEFI cases related to HBV vaccination occurred in men and women, respectively. The difference in AEFI cases due to HBV vaccination between men and women was similar to all vaccines in Guangdong Province. Among the 36,776 AEFI events reported during 2005–2014, 22,111 and 14,665 (1.51:1) AEFI cases reported by all vaccination occurred in men and women, respectively.

The Vaccine Adverse Event Reporting System (VAERS) in the United States reported 20,231 adverse events following the single (4444) and combination (15,787) uses of HBV vaccines from 2005 to 2015. Of the adverse events following the single HBV vaccination, 240 (5.40%) cases were infants aged less than 1 month, 235 (5.29%) were infants aged between 1.5 and 23 months, 310 (6.98%) were children aged between 2 and 18 years, and 2365 (53%) were adults older than 18 years of age. However, the data on age in the remaining cases (n = 1294, 29%) were unknown. Among the 15,787 adverse events following the co-administered vaccination, 9816 (62.17%) cases were infants aged between 1.5 and 23 months, 2278 (14.43%) were children aged between 2 and 18 years, and 3502 (22.18%) were adults older than 18 years of age. However, the data on age in the remaining cases (n = 191, 1.9%) were not reported in this study [27].

According to the national findings of HBV vaccination in China from 2005 to 2009, the incidence of AEFI and rare vaccine reaction were reported at 16.17 and 2.89 per million doses, respectively, which were lower than that reported in the present study. The incidence of AEFI and rare vaccine reaction reported by HBV vaccination in Jiangsu Province of China between 2007 and 2013 were reported at 164.09 and 17.39 per million doses, respectively, which were higher than that reported in this study [28]. Moreover, the incidence of AEFI and rare vaccine reactions related to HBV vaccination in Fujian Province of China during 2008 to 2013 were reported at 38.90 and 6.09 per million doses, respectively, which were close to the incidence reported in our study [29]. It should be noted that the varied cases in the major cities are attributed to the differences in the sensitivity of the monitoring system. As a long-term monitoring pilot province, the sensitivity of the AEFI monitoring system in Guangdong Province is higher than the national average one and lower than that in Jiangsu Province, which was another pilot province.

Internationally, a study from the United States reported 32,559 AEFI cases related to HBV vaccination between 1991 and 2001, with an incidence of 11.8 per 100,000 doses [30]. From 1 January 2000 to 30 September 2002, a study from Australia reported 145 AEFI cases related to HBV vaccination. The incidence rate was estimated at 10.4 per 100,000 doses [31]. In Brazil, a vaccine adverse events passive surveillance system was coordinated by the National Immunization Program 1999–2001, where three AEFI cases related to HBV vaccination were reported, with an incidence rate of 1.3 per 100,000 doses [32]. Additionally, 1016 AEFI cases related to HBV vaccination were observed in Cuba from 1999 to 2008, with an incidence of 20.9 per 100,000 doses [33].Our study reported the incidence of hep B vaccine AEFI was 3.54/100,000, lower than that reported in the United States, Australia, Brazil and Cuba. Biao Guo found in a 2013 system analysis that compared other vaccines that the HBV vaccine reported AEFI rates from Australia, Brazil, China, and the USA demonstrated the lowest levels [34].

Common vaccine reactions were the primary AEFI resulting from HBV vaccination, which accounted for 56.05%. More specifically, the most frequent clinical symptoms of common vaccine reactions were fever, redness, and swelling at the injection site. In contrast, most rare vaccine reactions were an allergic rash, which occurred in 716 cases with an incidence of 8.13 per million doses. Serious rare vaccine reactions were composed of thrombocytopenic purpura (n = 21), Henoch–Scholein purpura (n = 14), febrile convulsions (n = 6), anaphylactic shock (n = 2), laryngeal edema (n = 1), and encephalitis and meningitis (n = 1), with the incidence of 0.24, 0.16, 0.07, 0.02, 0.01, and 0.01 per million doses, respectively. These values indicated that the incidence of severe rare vaccine reactions in the present study were extremely low. As mentioned above, the VAERS in the United States had reported 20,231 adverse cases following the single HBV vaccination between 2005 and 2015. Of these, the most frequent adverse events were incorrect product storage (22%), and others included dizziness (8%), nausea (8%), fever (7%), headache (7%), rash (6%), pruritus (5%), and urticaria (5%). In regard to the adverse events following the combined use of the HBV vaccine and other vaccines, the most common manifestation was fever (23%), followed by injection site erythema (11%), vomiting (10%), irritability (10%), crying (9%), rash (8%), diarrhea (7%), injection site swelling (7%), erythema (7%), and urticaria (5%). Despite the different surveillance systems between the United States and China, the AEFI cases in the present study demonstrated similarities with that reported in the United States from 2005 to 2015.

According to the national monitoring results of adverse reactions to HBV vaccination in China from 2005 to 2009, the incidence of AEFI related to HBV vaccination was reported at 14.90 per million doses, and the values on common and rare vaccine reactions were reported at 12.01 and 2.89 per million doses, respectively [34]. In comparison, the incidence of AEFI, common vaccine reaction, and rare vaccine reaction that resulted from HBV vaccination in Guangdong Province were higher that at the national level. Among the rare vaccine reactions, the incidence of allergic rash was reported at 8.13 per million doses in the Guangdong Province, which was significantly higher than the national value (1.76 per million doses). The Chinese monitoring client report management system was launched nationwide from 2005 to 2006, and the MOH carried out an AEFI monitoring pilot in ten provinces of China. The monitoring results indicated that the reported cases varied dramatically between different regions. Of these, Guangdong, Jiangsu, Shanghai, Zhejiang, Hebei, and other pilot provinces reported the majority of cases, which accounted for 77% [35]. Following the national disease prevention and control information system platform, the Chinese national AEFI monitoring network direct reporting management system was discovered in 2008. Since then, 90% of the provinces in China have reported the AEFI cases. However, there was still a considerable difference in AEFI cases among varied regions [36]. As one of the pilot provinces in China, AEFI monitoring has been carried out in the Guangdong Province since 2005.

Our findings also showed that the incidence of AEFI and rare vaccine reactions related to HBV vaccination were lower than that of all vaccinations in the Guangdong Province, with the AEFI and rare vaccine reactions related to HBV incidence being 35.39 and 10.75 per million doses, respectively. According to the AEFI surveillance results of all vaccines in the Guangdong Province from 2005 to 2014, the incidence of AEFI and rare vaccine reactions was 81.76 and 21.85 per million doses, respectively.

The clinical symptoms of adverse events related to HBV vaccination were similar to those due to other vaccines. Moreover, adverse events with serious consequences are relatively rare. Anaphylactic shock represents a severe symptom of AEFI cases, which occurs rapidly following the vaccination and can lead to severe consequences without recovering in time. Two cases of anaphylactic shock were reported in this study. In comparison, the VAERS in the United States reported 37 cases of anaphylaxis from 2005 to 2015. Of these, 9 and 28 cases occurred following the single and combined use of vaccines, respectively. In light of the estimation from World Health Organization (WHO), the incidence of anaphylactic shock following the HBV vaccination was reported at 1.11–1.67 per million doses [12,13]. Our study found that the incidence of anaphylactic shock related to HBV vaccination (0.02 per million doses) was much lower than that announced from the WHO.

Limitations existed in the present study. First, this surveillance was conducted based upon a spontaneous reporting system; thus, the events were reported by the vaccination unit, with many cases being underreported. Second, the clinical diagnosis and classification of AEFI cases also depend on the ability of the reporting unit, which is quite different. Finally, we failed to compare the incidence of adverse events related to different types of HBV vaccines due to the lack of detailed information on different types of HBV vaccines applied in specific populations.

## 5. Conclusions

In conclusion, the reported adverse events and rare vaccine reactions related to HBV vaccination in the Guangdong Province occur within the expected range. Ongoing AEFI surveillance and scientific evidence responding to the public concerns about vaccine safety will help maintain public confidence in the vaccine and high HBV vaccination coverage.

## Figures and Tables

**Figure 1 vaccines-10-01087-f001:**
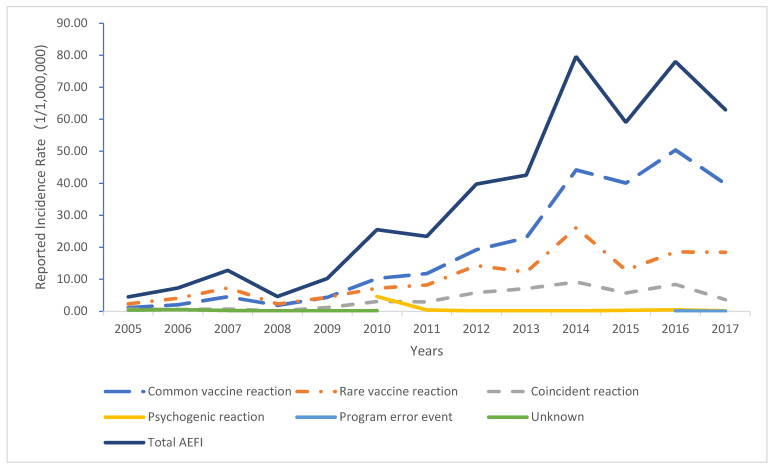
The incidence of adverse events related to HBV vaccination in Guangdong Province from 2005 to 2017 (1/1,000,000).

**Table 1 vaccines-10-01087-t001:** Number of AEFI cases related to HBV vaccination in Guangdong Province from 2005 to 2017.

Classification	2005	2006	2007	2008	2009	2010	2011	2012	2013	2014	2015	2016	2017	Total
Common vaccine reaction	7	11	20	13	29	104	92	125	165	289	273	323	295	1746
Rare vaccine reaction	14	22	32	16	29	73	64	93	88	171	88	119	137	946
Coincident reaction	4	3	3	1	8	31	23	38	51	60	39	54	27	342
Psychogenic reaction	0	0	0	1	0	47	3	1	1	1	2	3	1	60
Program error event	0	0	0	0	1	1	0	0	0	0	0	1	0	3
Unknown	2	3	1	1	1	2	0	1	0	0	0	0	0	11
Total	27	39	56	32	68	258	182	258	305	521	402	500	467	3115
Doses of HBV vaccine (10 thousand)	596.83	535.42	437.62	694.59	663.55	1012.06	777.93	648.67	717.6	654.4	680.96	641.03	741.65	8802.31

**Table 2 vaccines-10-01087-t002:** The adverse events related to HBV vaccination in Guangdong Province from 2005 to 2017 by gender and age.

Characteristic	2005	2006	2007	2008	2009	2010	2011	2012	2013	2014	2015	2016	2017	Total
Gender														
Male	17	26	32	18	32	133	114	147	172	288	249	296	277	1801
Female	10	13	24	14	36	125	68	111	133	233	153	204	190	1314
Age groups (year)														
<1	20	21	30	13	47	99	124	196	243	392	295	364	358	2202
1~<2	0	0	0	0	2	3	5	7	9	20	12	24	22	104
2~<3	2	3	4	1	2	4	2	6	6	8	10	8	14	70
3~<4	1	1	6	3	4	6	6	10	15	20	13	29	18	132
4~<5	1	4	5	5	0	0	7	8	10	21	25	16	14	116
5~<6	-	2	5	4	4	2	5	4	8	13	14	24	6	91
6~<7	1	2	0	0	0	5	2	3	5	18	13	8	10	67
≥7	2	6	6	6	9	139	31	24	9	29	20	27	25	333
Total	27	39	56	32	68	258	182	258	305	521	402	500	467	3115

**Table 3 vaccines-10-01087-t003:** Clinical symptoms of the adverse events related to HBV vaccination in Guangdong Province from 2005 to 2017.

Classification	Clinical Symptoms	Number of Cases	Incidence (Per Million Doses)
Common vaccine reaction	Fever	1046	11.88
	Redness and swelling	545	6.19
	Induration	63	0.72
	Others	72	0.82
	Vomiting	12	0.14
	Dizziness	3	0.03
	Diarrhea	3	0.03
	Nausea	1	0.01
	Headache	1	0.01
	Subtotal	1746	19.84
Rare vaccine reaction	Anaphylactic rash	716	8.13
	Urticarial	69	0.78
	Sterile abscess	31	0.35
	Angioedema	24	0.27
	Thrombocytopenic purpura	21	0.24
	Henoch–Scholein purpura	14	0.16
	Maculopapular rash	20	0.23
	Localized abscess	6	0.07
	Febrile convulsion	6	0.07
	Arthus reaction	2	0.02
	Lymphadenitis and lymphadenitis	2	0.02
	Anaphylaxis shock	2	0.02
	Other allergic reactions	16	0.18
	Laryngeal edema	1	0.01
	Brachial neuritis	1	0.01
	Encephalopathy	1	0.01
	Encephalitis and meningitis	1	0.01
	Others	13	0.15
	Subtotal	946	10.75
Total		2692	30.58

## Data Availability

Access to the data presented in this study can be acquired by connecting to the corresponding authors via email. The data are not publicly available due to restrictions of privacy.

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
