# Peer review of "Adverse Events of Vaccination against Hepatitis B Virus in Post-Marketing Surveillance from 2005 to 2017 in Guangdong Province, China"

_vaccines, 2022, doi:10.3390/vaccines10071087_

Round 1
Reviewer 1 Report
Respons HB
The article explains the reports of possible adverse reactions to the HB vaccine, at a time of crisis, and concluding that the safety profile is good. The article Is an opportunity to know the safety of vaccines in post-authorisation.
1. In the keywords remove the word: Introduction.
Introduction
2. Bibliographic reference 13 is from 1996, is there a more current one available?
3. Although there is talk of low endemia, other sources speak of medium situation. It is necessary to verify this aspect and justify it. I add bibliography:
https://bmcinfectdis.biomedcentral.com/articles/10.1186/s12879-019-4428-y
Liu J, Zhang S, Wang Q, Shen H, Zhang M, Zhang Y, et al. Lancet Infect Dis . 2016. gener; 16 ( 1 ): 80–6.
Cui F, Shen L, Li L, Wang H, Wang F, Bi S, et al. Emerg Infect Dis . 2017. maig; 23 ( 5 ): 765–72. 10.3201/eid2305.161477
https://www.ncbi.nlm.nih.gov/pmc/articles/PMC6453311/
4. Regarding vaccination coverage, Is there any more current data for vaccination coverage than in 2014?
5. Could reference 18 not be correct? Justify this point with other bibliographical references
6. The sentence: “There is a high likelihood of death caused by coincidental diseases following immunization of HBV vaccine among children aged less than 1 month” justify it.
7. 17 cases of deaths + 6 deaths in Guandong? What is the total?
Methods
8. HBV vaccine : I think it is important for public knowledge to add a table with the differences between the vaccines. And also add if these were always available throughout the studied time. I think a vaccine analysis is important.
When these vaccines are given they are given along with others. Is there co-administration?
Report System:
9. Add that it is a passive pharmacovigilance system.
10. Who can notify? Citizens too? Is there a system of financial reward for an adverse reaction? These points should appear in the text.
Report scope
11. The classification: “Regarding AEFI, it can be categorized into common vaccine reaction, rare vaccine reaction, coinci-dent reaction, psychogenic reaction, program error event, and unknown.” Follow some bibliographical reference, for example from the WHO. Add this.
12. Did the common reports coincided with those described in the EPAR? Did all the vaccines administered have the same safety profile? Describe this point.
13. Distribute table 1 so that it is not cut. This point is for the publisher.
14. In the title: “Table 1. AEFI cases caused by HBV vaccination in Guangdong from 2005 to 2017.”
Be careful with words since causality analysis has been performed and they are caused or are they notified? Clarify this point. Correct it on all tables.
15. Is there any explanation for the increase in notifications? Type of vaccine?
16. Can you describe the errors reported?
17. If the problem of sudden deaths appeared in 2014, an increase in the reporting sensitivity is observed since 2009. Could there be any factor that explains this fact? This increase is seen in all types of described reaction type, except the psychogenic reaction , which in 2010 was widely reported. What was this fact due to?
18. In the table 3. Are there any factors that explain the reports cases in children aged ≥ 7 years? What are the most frequent types of reactions reported in this age group? Describe possibilities for this as vaccination is carried out at 0, 1 and 6 months of age.
Discussion
19. In the sentence : “Our find-ings indicated that the lowest incidence of AEFI caused by HBV vaccination was” is caused by the vaccine or reported/notified? Look out and clarify these terms.
In: “the lowest incidence was observed in 2005 at 2.35 per million doses” are also notified.
20. The sentence: “Following the national disease prevention and control information system platform, the Chinese national AEFI monitoring network direct reporting management system was discovered in 2008. Since then, 90% of provinces in China have reported” As it is from 2008, this fact has no implication on the sensitivity of the register to explain the increase in notifications from 2010 onwards. Can you elaborate on this point?Conclusion
21. Is only one vaccine used? Or different vaccines?
Reviewer 2 Report
Overview and general recommendation: the authors conducted a study of surveillance of AEFI events in more than 88 million doses of HBV vaccines from 2005 to 2017 in Guangdong Province, China. The conclusion is that the incidence of AEFI events is extremely low (35.39 per million doses). Overall, the topic is interesting and important, but the presentation and interpretation of the study need to be greatly improved. I have the following comments for the authors’ consideration.
Major comments:
1. My primary concern about this study is the interpretation of data. Although the authors compiled massive data derived from a large group over a decade, I cannot find any comprehensive or systematic analysis of these data between different collection years or different patient groups. At least, it is recommended to move the comparative analysis of AEFI between Guangdong Provinces and other Provinces in China from the Discussion section to the Results section.
2. In the Abstract section: please unify the decimal places of six percentages. Additionally, the reviewer suggests adding the specific numbers after all the percentages to avoid confusion, e.g., 74.01% (XX/XXX).
3. Introduction section, last paragraph: the authors should clarify if HBV vaccination is not responsible for the 18 severe adverse events and 17 death cases, what is the causation then?
4. The Data analysis section could be combined into the Methods section. The authors mentioned five HBV vaccines are available in China in section 2.1. Are there any remarkable differences in AEFI from the distinct five HBV vaccines? Please specify.
5. Result section: the numbers in the Tables 1–3 are difficult to comprehend. The reviewer highly suggests using histograms instead of Tables for a better presentation. Paragraph 1, line 7: the “upward trend annually” seems incorrect, the incidence rather fluctuating.
6. The discussion section, paragraph 2, line 6: I am confused about the “36,776 AEFI events due to HBV vaccination in Guangdong Province between 2005 and 2014”. What are the 3115 AEFI events between 2005 and 2017 in this study then? Paragraph 3, line 6: please specify “other vaccines” and provide relevant references.
7. The discussion section, the authors should provide supporting references in paragraphs 6 and 7. Notably, be cautious of the statement “the HBV vaccine is safer than that in all vaccines”.
8. The Discussion section is very long, and many sentences can be moved to the Results section, especially the description of comparisons of AEFI in Guangdong Province and Jiangsu Province, and the United States. For example, paragraphs 7 and 8.
Minor comments:
1. Abstract, line 4: please define “AEFI”.
2. Keywords: delete “Introduction”.
3. Methods 2.3.: please define “MOH” and “SFDA”.
4. Discussion, paragraph 5, line 2: change “Chine” to “China”.
5. In the last paragraph of the Discussion section, delete the sentence “we performed the AEFI monitoring for HBV vaccination from 2005 to 2017, covering a vaccination dose of more than 88 million. However,”.
6. Reference: the references 12 and 16 are identical. Please use proper citation software and double-check the references carefully.
Reviewer 3 Report
Safety Profile of Vaccination against Hepatitis B virus in Postmarketing Surveillance from 2005 to 2017 in Guangdong, China
Vaccine
Thank you for asking me to review the above-titled manuscript. The title is intriguing. However, there are significant problems with the manuscript.
- Title: Is the study about (Safety Profile)? or (Adverse effects)? These two are not the same. What does "postmarketing" mean?
- Abstract- 1) Which Hepatitis B vaccine? State full name and licence, manufacturer. 2) What is the source of data? 3) Why this period 2005 to 2017 in particular. 4) How data were collected and analysed? 5) Were these neurological deficits permanent or temporary? 6) How are these figures differ from those reported in the USA, Canada, the UK, and Australia. The conclusion should be based on a comparison to other standards/findings.
- Introduction - 1) When was the Hepatitis B vaccine applied widespread in China? 2) Is this the same vaccine used in the USA, Canada, Europe and Australia? 3) What did trigger the study? 4) What is the research question? 5) Are you after "Adverse effects", or "Quality" or "Safety"? These are different issues.
- The abstract is about the "adverse effects." , the title is about "Safety", the aims of the study about "Adverse effects", "Quality" and "Safety", Tables about "Adverse Effects" - Which one should we believe?
- Methods- 1) Was the licence to manufacture the vaccine in China from GlaxoSmithKline and Merk? State this clearly. 2) The study needs ethical approval. 3) Items 2.2 and 2.3 are not clearly written. How these are methods? 4) How the Report Scope was used and applied? 5) What is the source of the data? Which database was searched? 6) How did the authors search the database? 7) Data Analysis (Item 3)- Where is the data analysis? What methods were used? 7) No methods or tools are described for "Safety" and "quality." - There is inconsistency in the paper.
- English editing of the whole manuscript is needed.
- Results- Tables do not show "Safety" no "Quality."
- Discussion- 1)The authors repeat mentioning their results. Is this the purpose of the discussion? 2) again, mixing "adverse effects" with "safety". Look for the headings of your tables and reads your results. 3) Compare your results against those reported from the USA, Canada, the UK, Europe, and Australia. 4) Discussion is poorly written.
- There are several other limitations in the method used in the study, we do not know the pattern of reporting and dits distribution, areas with low reporting versus areas with high reporting. No report about quality measures taken to ensure standardised reporting.
- Conclusions- You cannot make any conclusions without comparing your findings against those reported internationally- in South East Asia, the USA, the UK, Canada, Europe, and Australia.
- References- 1) Many references are from Chinese journals- For example numbers, 17, 18, 21, 22, 24, 27, 28, 29, 30 and others. How can the Editor, Peer-Reviewers and readers check on these journals? 2) Some references are too old—for example, 13 and 20. 3) Important re
Round 2
Reviewer 1 Report
I encourage the authors to make some tweaks to improve the manuscript.
Introduction
1 Why is there only data on vaccination coverage of the first dose for the year 2015-2017?
2. The sentence: To address this issue, we conducted the surveillance of adverse events related to HBV immunization between 2005 and 2017 to investigate the quality and safety of the HBV vaccine in Guangdong Prov-ince, China. Is not apropiated becouse the study is de adverse events and not the quality of HB vaccines.
Methods
3. Although the information on the available vaccines has been improved, we do not know if any of them are adjuvant and which is the adjuvant. These vaccines are not authorized in Europe, except of the GSK.
4. Following the previous point, the pharmacovigilance systems must have as essential information the drug to which the clinic that has triggered the notification is attributed, in fact in the pharmacovigilance regulations it is essential information. You have not commented on the option of carrying out an analysis by type of vaccine. This analysis is important. In the event that it cannot be done, it would be necessary to explain it very well and describe said limitation.
5. Add the nombre of the ethics committee: The research protocol was approved by the ethics committee of Southern Medical University.
Result
6. In the title of table 1 is the same Guangdong and Guangdong Province?
7. I don’t think it is appropriate for data not worked as results to appear in the results section, specifically those of the references of 25, 26, 27, 28. It would be more of a discussion.
8. It is necessary to adapt the bibliography to the standards of the journal. review them all.
Author Response
Dear reviewer,
Thank you very much for your concern on our manuscript titled “Adverse Events of Vaccination against Hepatitis B virus in Post-marketing Surveillance from 2005 to 2017 in Guangdong Province, China”. The reviewers’ and your professional suggestions and comments are vital to enhancing the manuscript’s quality and our future work. We have now completed the minor revision based on the comments and tried our best to improve the manuscript. Please see our point-by-point responses as listed below.
We appreciate your warm work earnestly and hope that the revisions will meet with approval.
Yours sincerely,
Qing Chen
Introduction
Point 1: Why is there only data on vaccination coverage of the first dose for the year 2015-2017?
Response 1: Thank you very much for your comment. In this study, we provided the data on HBV vaccination coverage of the first dose according to a systematic review and meta-analysis focused on HBV infection in the general population of China. However, the reasons for vaccination coverage of the first dose for the year 2015-2017 were not mentioned. To avoid any confusion, we have removed this sentence from the revised manuscript.
Point 2: The sentence: To address this issue, we conducted the surveillance of adverse events related to HBV immunization between 2005 and 2017 to investigate the quality and safety of the HBV vaccine in Guangdong Province, China. Is not appropriated because the study is de adverse events and not the quality of HBV vaccines.
Response 2: Thank you for your indication. We are sorry that we failed to correct this sentence during the first revision. Now we have changed it to the sentence “To address this issue, we conducted the surveillance of adverse events related to HBV immunization between 2005 and 2017 to investigate the adverse events in relation to the HBV vaccination in Guangdong Province, China” in the revised manuscript. (Page 2)
Methods
Point 3: Although the information on the available vaccines has been improved, we do not know if any of them are adjuvant and which is the adjuvant. These vaccines are not authorized in Europe, except of the GSK.
Response 3: Thank you for your comment. As mentioned in the manuscript, there are five recombinant HBV surface antigen vaccines have been available for the public from 2005 to 2017 in Guangdong Province of China. Of these, four HBV vaccines are made in China, whose production standard has obtained the regulatory certification from the World Health Organization. Moreover, the adjuvant used in the Chinese production are aluminum hydroxide and sodium chloride et al., which are same as the vaccine manufacturing by GlaxoSmithKline Biologicals, U.K.-licensed. We have now included these statements in the revised manuscript. (Page 2)
Point 4: Following the previous point, the pharmacovigilance systems must have as essential information the drug to which the clinic that has triggered the notification is attributed, in fact in the pharmacovigilance regulations it is essential information. You have not commented on the option of carrying out an analysis by type of vaccine. This analysis is important. In the event that it cannot be done, it would be necessary to explain it very well and describe said limitation.
Response 4: Many thanks for your comment. We are sorry that we did not addressed the option of conducting an analysis by types of vaccines. Due to the lack of detailed information on the different types of HBV vaccines applied in specific populations, we failed to compare the incidence of adverse events related to different types of HBV vaccines in the present study. Therefore, we have described it as one of the limitations in the revised manuscript. (Page 11-12)
Point 5: Add the number of the ethics committee: The research protocol was approved by the ethics committee of Southern Medical University.
Response 5: Thank you very much for your kindly reminder. The number of the ethics approval was not currently available. Instead, we have provided the copy of ethics statement.
Result
Point 6: In the title of table 1 is the same Guangdong and Guangdong Province?
Response 6: Many thanks for your carefully reading. It is the same for Guangdong and Guangdong Province. To be consistent, we use “Guangdong Province” in Table 1. (Page 5)
Point 7: I don’t think it is appropriate for data not worked as results to appear in the results section, specifically those of the references of 25, 26, 27, 28. It would be more of a discussion.
Response 7: Thank you very much for your comment. We have removed them from the Results section to the Discussion section.
Point 8: It is necessary to adapt the bibliography to the standards of the journal. review them all.
Response 8: Many thanks for the suggestion. We have reviewed and checked the bibliographies based on the standards of the journal.
Reviewer 2 Report
The authors have adequately addressed my concerns and comments in their revised manuscript.
Author Response
Thank you very much for your concern on our manuscript. Your professional comments play an important role in improving the manuscript’s quality.